# Subsurface Detection of Shallow Targets by Undersampled Multifrequency Data and a Non-Cooperative Source

**Adriana Brancaccio \*, Angela Dell'Aversano, Giovanni Leone and Raffaele Solimene**

Dipartimento di Ingegneria, Università della Campania Luigi Vanvitelli, via Roma 29, 81031 Aversa, Italy; angela.dellaversano@gmail.com (A.D.); giovanni.leone@unicampania.it (G.L.); raffaele.solimene@unicampania.it (R.S.)

\* Correspondence: adriana.brancaccio@unicampania.it; Tel.: +39-081-5010270

**Abstract:** Imaging buried objects embedded within electrically large investigation domains can require a large number of measurement points. This is impractical if long data acquisition time cannot be tolerated or the system is conceived to work at some stand-off distance from the air/soil interface; for example, if it is mounted over some flying platform. In order to reduce the number of spatial measurements, here, we propose a method for detecting and localizing shallowly buried scattering targets from under-sampled far-field data. The method is based on a scattering model derived from the equivalence theorem for electromagnetic radiation. It exploits multi-frequency data and does not require that the transmitter and receivers are synchronized, making the source non-cooperative. To provide a benchmark against which spatial data have to be reduced, first, the number of required spatial measurements is examined by analyzing the properties of the relevant scattering operator. Then, since under-sampling data produces aliasing artifacts, frequency diversity (i.e., multi-frequency data) is exploited to mitigate those artifacts. In particular, single-frequency reconstructions are properly fused and a criterion for selecting the frequencies to be used is provided. Numerical examples show that the method allows for satisfactory target transverse localization with a number of measurements that are much less than the ones required by other methods commonly used in subsurface imaging.

**Keywords:** electromagnetic inverse scattering; microwave imaging; non-cooperative sources; shallow subsurface prospecting

---

## 1. Introduction

The problem of detecting buried targets through electromagnetic waves is a classical inverse scattering problem which is relevant in a number of different applications [1,2] such as civil engineering diagnostics [3–9], archaeological and geophysical prospecting [10,11], cultural heritage monitoring [12,13], mines and IEDs (improvised explosive devices) detection [14], only to mention a few of them. Electromagnetic waves at the microwave spectrum allow to get high resolution. That is why over the years a great deal of research has been carried out to tackle the many challenges of subsurface imaging, both in terms of the system and the imaging algorithms [15–21].

Ground Penetrating Radar (GPR) systems generally work in contact with the air/soil interface. More recently, research has been considering GPRs mounted over airborne platforms as well [22,23]. This is because a flying GPR can offer the great advantage of inspecting large areas quickly. However, in this framework, the necessity to properly trade-off the system cost and the achievable performance becomes even (with respect to the usual on the ground GPRs) more critical.



The most common GPR sensor arrangement is by far the multi-monostatic configuration, where each sensor (or even a single moving sensor) acts simultaneously as transmitter (TX) and receiver (RX). The point is that for long measurement lines (which are needed to obtain large synthetic apertures in order to obtain high transverse resolution) and large investigated areas, the multi-monostatic arrangement soon requires a huge number of measurement points. These can be achieved by a multiple-pass (over the scene under test) strategy but this clearly hinders fast imaging. Alternatively, a fleet of sensors can be used. In this case, the system soon becomes too expensive. Therefore, the need for a reduced number of sensors, that in turn should also be relatively cheap, arises.

In this regard, a single-view/multistatic configuration seems convenient. Indeed, in this case, only one sensor transmits, whereas the others act like mere receivers. Therefore, they have reduced weight and can be mounted over simple and cheap flying platforms. Of course, synchronization between TX and RX is much more difficult than in the multi-monostatic arrangement, where TX and RX are co-located and share the same electronic system. In order to address this drawback one can resort to some methods used in passive radar technology. In those cases, the receivers estimate the TX parameters, such as the timing, the source position and sometimes also the transmitted waveform, provided that a direct link between TX and RX exists [24,25]. However, apart from the required additional processing, in this approach the direct link can blind the radar since the field coming from the targets is generally much lower than the direct one, that could saturate the receiver dynamic.

The present paper is a development of the approach already presented in [26], where we adopted a different point of view and introduced a reconstruction scheme that allows to avoid the need for synchronization and, hence, the direct link blind effect as well. This approach is strictly connected to the particular scenario we are interested in, where the targets are assumed buried in close proximity to the air/soil interface (i.e., a few centimeters beneath the air/soil interface just to hide them from sight), as for mine [27] or unexploded improvised device [28] detection. In this case, depth information can be disregarded and the main goal is to detect and transversely localize the targets. Accordingly, frequency band is not necessary and single-frequency detection is possible. However, it is assumed that the receivers simultaneously collect the scattered field or at least that the measurement stage is completed within the source coherence interval.

According to [26], the scattered field is modeled as being radiated by equivalent surface currents supported at the air/soil interface (or at some reference plane above the target), which indeed retain information about the existence and the transverse location of the buried targets. Moreover, imaging is cast as a 2D inverse problem which is computationally convenient. It is remarked that the system can be considered to operate like a passive radar. However, while the source is actually non-cooperative (in the sense that it does not share any information with the receivers), it is not opportunistic since it is deliberately deployed in the scene.

The proposed model assumes that the equivalent current radiates in free-space so that a free-space propagator is considered while achieving the reconstructions. This assumption is not so critical since the targets are shallowly buried. In other words, the propagation path in free-space is much longer than the one in the half-space. Also, the proposed method can experience the drawback that targets, which are not close to the chosen reference plane, can be difficult to detect. To avoid this problem, reconstructions over plane at different depths are computed.

Even under the above considered framework, the number of required sensors soon become unfeasible when the area to be investigated is even only moderately large (in terms of wavelength). This question is clearly related to the so-called number of degrees of freedom (NDF) [29] of the pertinent scattering operator. Accordingly, the first contribution we give in this paper is the estimation of the relevant NDF. As we will show, collecting the data according to the NDF is practically impossible. Consequently, reconstructions will unavoidably be aliased. Therefore, the second contribution concerns a simple multi-frequency procedure to mitigate aliasing artifacts. More in detail, since the TX and the RXs are not synchronized, multi-frequency data cannot be processed coherently (simultaneously). Hence, each frequency data is processed separately. Of course, each single-frequency reconstruction

results corrupted by aliasing artifacts. However, since such artifacts are frequency dependent they corrupt single-frequency reconstructions in a different way. This feature is then exploited to mitigate such artifacts by devising a simple scheme which fuses the different single-frequency reconstructions.

The paper is organized as follows. In Section 2 the scattering model introduced in [26] is briefly recalled. Section 3 is devoted to the study of the properties of the scattering operator in order to find the NDF and to highlight how the number and the positions of the receiving sensors affect the spatial resolution. In Section 4, we introduce a technique that can mitigate aliasing artifacts by properly processing single-frequency images. In addition, a criterion for choosing the working frequencies is provided. Finally, numerical examples are reported in Section 5.

## 2. Surface Scattering Model

In this section we briefly recall the relevant scattering equations according to the model presented in [26].

Consider a half-space background medium with the separation interface (i.e., the air/soil interface) at $z = 0$. The scattering targets to be detected are assumed buried in the lower half-space (i.e., for $z < 0$) and located within the investigation domain $D = [-x_{in}, x_{in}] \times [-y_{in}, y_{in}] \times [z_{in}, 0]$, with $z_{in} < 0$. The electromagnetic features of the soil are unknown.

The scattering scene is probed by a single source located at some stand-off distance (from the air/soil interface) from the air side. Its location, timing and radiated waveform are not known. What matters for our detection scheme is only the working frequency band. The field scattered by the buried targets is collected over a set of sensors still located on the air side and all at height $h$ above the ground. Therefore, all sensors belong to a given measurement plane: say $\mathbf{r}_n$, with $n = 1, 2, \cdots, N$ their positions. Hence, a single-view/multistatic configuration is considered. The sensors are assumed to be in the far-zone of the scene under investigation. Figure 1 shows a pictorial view of the scattering configuration and illustrates the adopted reference frame.

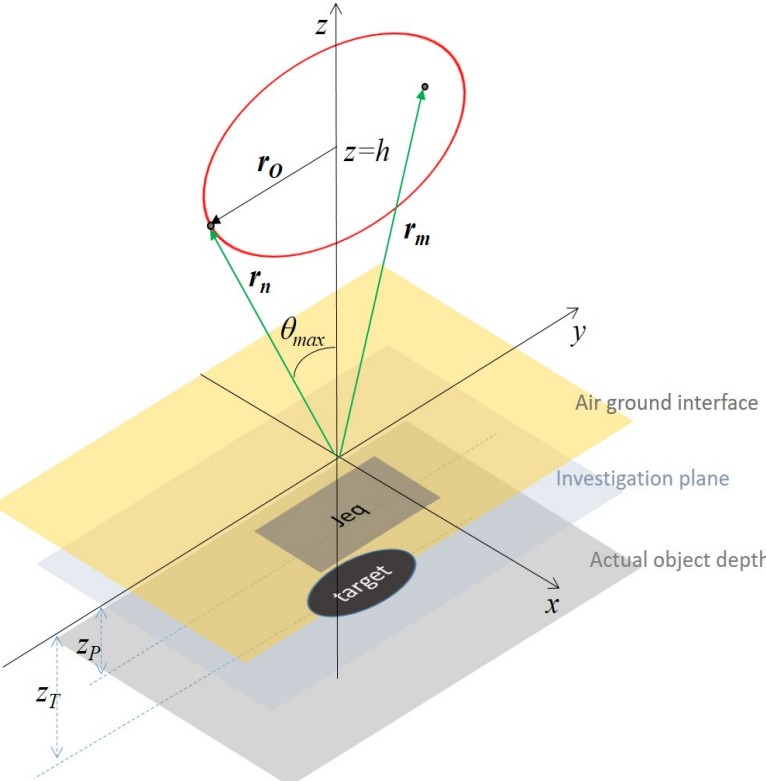

**Figure 1.** Pictorial view of the scattering configuration and reference system. The air/soil interface is at $z = 0$, the scattering object depth is $z_T < 0$ and the equivalent plane at $z_p$.

Say the target is buried at the unknown depth $z_T \in [z_{in}, 0]$. According to the equivalence theorem [30], its scattered field can be considered as being radiated by equivalent surface currents, both electric and magnetic, just supported over a plane at $z_p > z_T$. We address this as the equivalent plane (see Figure 1). By filling the half-space $z < z_p$ with a perfect magnetic conductor, only the electric current turns out to be relevant. Moreover, the magnetic conductor can be removed by doubling the electric current. Accordingly, under the far-zone approximation and a proper transformation of the far-field components, the scattered field can be expressed as [26]

$$\begin{aligned}
\bar{\mathbf{E}}_S(\phi, \theta) = \zeta k^2 \exp\left(jkz_p \cos\theta\right) \\
\int_{D_T} \exp\left[jk \sin\theta(x\cos\phi + y\sin\phi)\right]\mathbf{J}_{eq}(x, y)dxdy
\end{aligned} \tag{1}$$

where $D_T = [-x_{in}, x_{in}] \times [-y_{in}, y_{in}]$ is the investigation domain cross section, the observation point **r** has been expressed in terms of its spherical components, i.e., $r$, $\theta$ and $\phi$, $\zeta$ and $k$ are the characteristic impedance and the wavenumber of the free-space.

Some explanations are in order.

First, we observe that, to be rigorous, surface integration should run over the entire plane $z = z_p$ since equivalent currents do so. However, the target is very close to the air/soil interface, and hence to the plane $z = z_p$, therefore, it is reasonable to assume that the current support is very similar to the target's cross section. Hence, $D_T$ needs to be only large enough to account for all the possible positions of the targets.

It must be clarified that the model in Equation (1) assumes a free-space Green function. This is clearly correct when $z_p = 0$, that is when the equivalent plane coincides with the air/soil interface. When this is not the case, the model is basically neglecting the part of propagation that takes place in the soil. The main effect of this approximation is that the targets act as if they were more deeply located (because soil is electromagnetically denser than air). However, this is not a serious drawback because the targets are shallowly buried, i.e., the propagation path in the half-space is much shorter than the one in free-space. What is more, choosing $z_p$ not necessarily equal to zero gives a degree of freedom which can be used to adjust focusing in the imaging stage. This allows, for example, to compensate for uncertainties of the sensor height, and, even more importantly, to scan the depth. The latter proves to be very useful if the scene consists of targets buried at different depths (as it is shown in the following).

Finally, it is remarked that the field in (1), $\bar{\mathbf{E}}_S$, is not directly the scattered field but it is related to it through a linear transformation, that is

$$\bar{\mathbf{E}}_S = -j \exp(-jkr)/(2\pi kr)\underline{\mathbf{T}}^{-1}\mathbf{E}_S \tag{2}$$

where $\mathbf{E}_S = (E_{xS}, E_{ys})$ is the scattered field projected in the $x - y$ plane and $\underline{\mathbf{T}}$ is defined as

$$\underline{\mathbf{T}} = \begin{bmatrix} \cos^2\theta\cos^2\phi + \sin^2\phi & -\sin^2\theta\cos\phi\sin\phi \\ -\sin^2\theta\cos\phi\sin\phi & \cos^2\theta\sin^2\phi + \cos^2\phi \end{bmatrix} \tag{3}$$

Note that $\underline{\mathbf{T}}$ is not singular (and, hence, invertible) since in practical cases one can collect the scattered field over an aspect limited configuration so that the singularity point $\theta = \pi/2$ is excluded.

The proposed model leads to a number of advantages. Imaging is cast as a 2D inverse problem and this reduces the computational burden as compared to a 3D formulation. Also, note that the vector problem is actually split into two identical scalar problems. Indeed, $J_{xeq}$ and $J_{yeq}$ can be separately determined from $\bar{E}_{xS}$ and $\bar{E}_{yS}$ by inverting the same integral operator. Moreover, as mentioned above, there is no need to know or estimate the soil properties. Of course, this is made possible as we renounced to estimate the target's depth.

Finally, we want to highlight why source information (or, equivalently, the incident field knowledge) is not necessary. This is because the incident field is embodied within the equivalent surface

currents and, hence, does not affect the operator to be inverted for the imaging. However, those sources are frequency dependent. Accordingly, frequency data cannot be simultaneously exploited to get the reconstructions. Rather, each frequency datum needs to be processed separately and eventually fused.

## 3. NDF Analysis and Space Sampling Cues

In order to obtain information on how to deploy the sensors, here we study the singular value decomposition of the described single-frequency scattering model.

Let us rewrite Equation (1) in the following way

$$
\bar{E}_S(u,v) = (2\pi)^2 \zeta \exp\left[jkz_p\sqrt{1-(u^2+v^2)}\right]
$$
$$
\int_{-\chi_{in}}^{\chi_{in}} \int_{-\eta_{in}}^{\eta_{in}} \exp\left(j2\pi u\chi\right) \exp\left(j2\pi v\eta\right) J_{eq}(\chi,\eta) d\chi d\eta \tag{4}
$$

where $u = \sin\theta\cos\phi$ and $v = \sin\theta\sin\phi$ are the observation variables and $\chi_{in} = x_{in}/\lambda$ and $\eta_{in} = y_{in}/\lambda$ are the dimensions of the investigation domain normalized to the wavelength $\lambda$. Still denote the normalized investigation domain as $D_T$. Note that here $\bar{E}_S$ can be one of the two components of the scattered field in Equation (1). Equation (4) defines a Fourier-like linear non-symmetric operator

$$
\mathcal{A} : J \in L^2(D_T) \rightarrow \bar{E}_S \in L^2(\Omega) \tag{5}
$$

with $L^2(D_T)$ and $L^2(\Omega)$ the sets of square integrable complex valued functions supported over $D_T$ and $\Omega$, $\Omega$ being the observation domain in the $(u,v)$ plane. Such an operator is compact [31]. Moreover, as the relevant kernel function is of exponential type [32], its singular values are expected to have an exponential fast decay beyond a certain index. Accordingly, this index can be identified as the NDF of the problem. The NDF actually depends also on $\Omega$ which in turn depends on the measurement aperture. Here, according to the previous choice of the observation variables, it seems natural to consider the measurements taken over a portion of the measurement plane, $\Sigma$, located at the height $h$ from the air/soil interface and enclosed within the circle of radius $r_O$, that is $\Sigma = \{(x,y,z) : z = h, x^2 + y^2 \le r_O^2\}$ or equivalently, $\Sigma = \{(r,\theta,\phi) : r = h/\cos\theta, 0 \le \theta \le \tan^{-1}(r_O/h), 0 \le \phi < 2\pi\}$. Accordingly, $\Omega_C = \{(u,v) : u^2 + v^2 \le \sin^2[\tan^{-1}(r_O/h)]\}$, where the subscript $C$ refers to the circular shape of the domain. Unfortunately, in this case, even though $\mathcal{A}$ has a simple expression, its singular system is not known in closed form.

To further elaborate on this question, we note that if $\Omega$ were rectangular, say $\Omega_R = [-U, U] \times [-U, U]$, and considering that the exponential factor outside the integral in (4) is a unitary transformation, the singular values of $\mathcal{A}$ would be actually known to be linked to the prolate spheroidal wave-functions [33]. Accordingly, the $\sigma_n(\Omega_R)$s would exhibit an abrupt decay roughly in correspondence of the index

$$
n = [(2U)^2 4\chi_{in}\eta_{in}] \tag{6}
$$

Moreover, it can be shown that if $\tilde{\Omega} \supset \Omega$ then $\sigma_n(\Omega) \le \sigma_n(\tilde{\Omega})$[34]. Accordingly, by considering two rectangular domains such that $\Omega_{Rout} \supset \Omega_C \supset \Omega_{Rin}$, it is easy to find upper and lower bounds for the NDF of $\mathcal{A}$ as

$$
8\sin^2\theta_{max}\chi_{in}\eta_{in} \le N_T \le 16\sin^2\theta_{max}\chi_{in}\eta_{in} \tag{7}
$$

where $\theta_{max} = \tan^{-1}(r_O/h)$. In Figure 2, the behavior of the lower bound of $N_T$ is reported as a function of the normalized dimension of the investigation domain, assumed square, for different values of the ratio $r_O/h$. As can be seen, the number of required data quickly becomes unfeasible as the size of the investigated area increases.

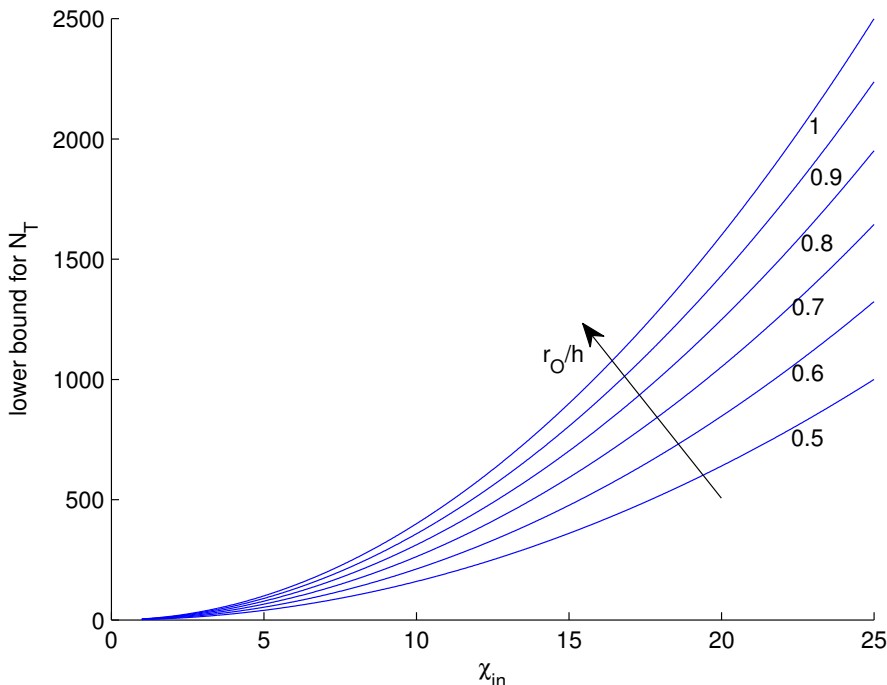

**Figure 2.** Illustrating the behavior of the lower bound of $N_T$ as a function of the investigation domain normalized side, for different values of the ratio $r_O/h$.

The properties of prolate spheroidal wave-functions [33] suggest sampling data in the $u - v$ plane over a regular grid at a step of

$$\Delta u = 1/(2\chi_{in})$$
$$\Delta v = 1/(2\eta_{in})$$
(8)

along $u$ and $v$, respectively [33]. If $(u_n, v_m)$ are the corresponding sampling points, with $u_n \in (u_1, u_2, \cdots, u_N)$, $v_m \in (v_1, v_2, \cdots, v_M)$ and $NM = N_T$, the sensors' positions in Cartesian coordinates can be obtained by the following transformations

$$x_{nm} = hu_n/\sqrt{1 - (u_n^2 + v_m^2)}$$
$$y_{nm} = hv_m/\sqrt{1 - (u_n^2 + v_m^2)}$$
(9)

These are non-linear transformations, therefore the uniform sampling in the $u - v$ plane translates into a non-uniform sampling in the $x - y$ plane. This is shown in Figure 3 (top panels). It is clear that this would lead to a quite unusual sensor arrangement. On the contrary, a uniform sampling in $x - y$ would result non-uniform in $u - v$. In Figure 3 (bottom panels) the relative behavior is shown. It can be concluded that in order to properly sample the field in the $u - v$ domain, either a non-uniform sampling can be adopted in the $x - y$ plane or a uniform sampling can be used, having care that Equation (8) is satisfied in the central part of the observation domain. In the latter case, $u - v$ data are over-sampled in the peripheral zone as is shown in Figure 3 (bottom panels).

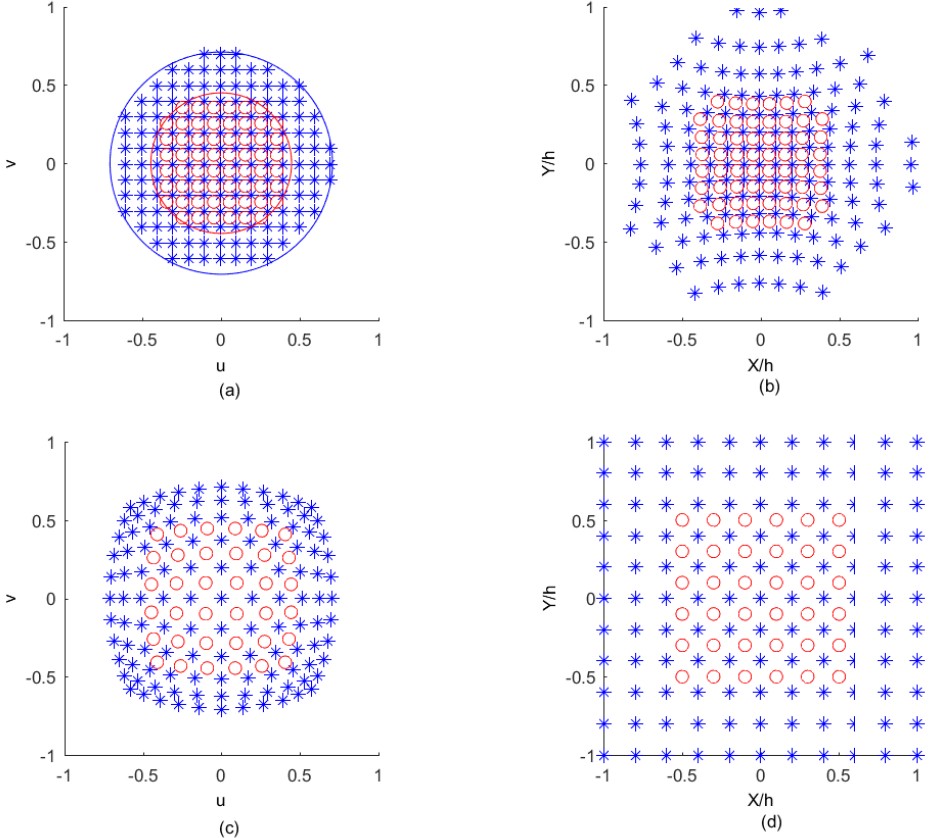

**Figure 3.** Relationship between the sampling points in the spectral (panels (**a**) and (**c**)) and spatial (panels (**b**) and (**d**)) domains for $\chi_{in} = 5$ and different measurement configurations. (**a**) shows the uniform sampling inside a circular $u - v$ domain for $r_O/h = 1$ (blue stars) and $r_O/h = 0.5$ (red circles), whereas the corresponding non-uniform sampling in the $x - y$ plane is reported in (**b**). As opposed, in (**d**) a uniform sampling is employed for a square $x - y$ domain of side $2X_{max} = 2h$ (blue stars) and $2X_{max} = h$ (red circles), whereas the corresponding points in the spectral domain are reported in (**c**).

The importance of choosing appropriate sampling points is illustrated in Figure 4, where the singular values corresponding to the even sampling in the $u - v$ domain are compared to those corresponding to equally spaced samples in the $x - y$ domain. The number of samples and the spatial observation region are the same in both cases. As can be seen, uniform sampling in the $u - v$ leads to a "flat" singular values behavior before the index provided by Equation (6) ($n = 200$ for the present case), by contrast, uniform sampling in $x - y$ leads to singular values that start decaying even before the critical index. This of course implies a different effect of noise on the inversion [35].

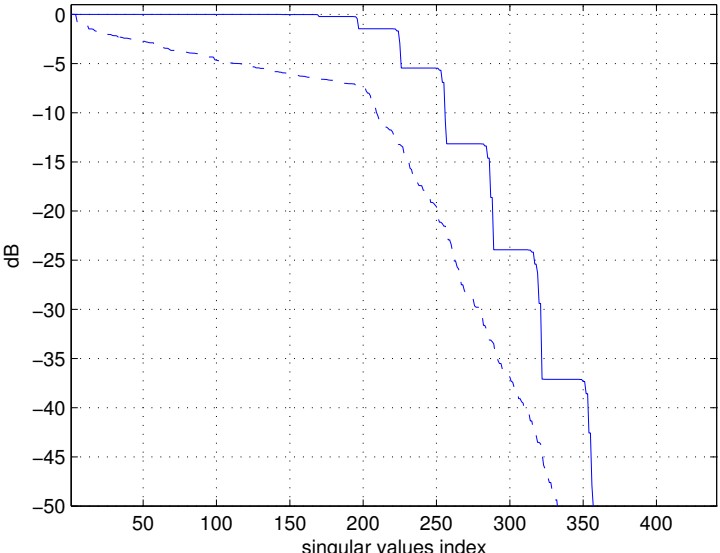

**Figure 4.** Singular values of the operator $\mathcal{A}$: $21 \times 21$ sampling points are taken over a square of side $2u_{max} = \sqrt{2}$ in the $u - v$ plane or $2X_{max} = 2h$ in the $x - y$ plane; $\chi_{in} = 5$. The solid line refers to the uniform sampling in $u - v$, the dashed line to the uniform sampling in $x - y$.

## 4. Multi-Frequency Method

In the previous section we showed that the number of data soon becomes huge for an electrically large investigation domain. Apart from the data storage requirements and the high computational burden, this would require an unfeasible sensing configuration or very long measurement time. To mitigate these problems, one is forced to work with a reduced (with respect to the NDF) number of data but aliasing then occurs. Since most electromagnetic sensors are able to acquire data at different frequencies at a relatively low cost, it appears useful to investigate whether it is possible to 'trade' space data for frequency data. In other words, we suggest to remedy the reduction of spatial measurements with frequency data [36].

According to our scattering model, since source information is unknown to the receivers, multiple frequency data cannot be processed simultaneously. Actually, each frequency needs to be processed separately. By single-frequency data, under-sampling gives rise to artifacts due to aliasing in the reconstructions. Therefore, each single-frequency under-sampled reconstruction will exhibit, besides the actual target reconstruction, also a number of replicas whose spatial positions, however, change with the frequency. This feature can be exploited in order to reduce the impact of the replicas by introducing a suitable 'fusion' strategy which combines the different single-frequency reconstructions. At this juncture we point out that the frequency fusion procedure cannot aim at correctly imaging the scattering objects, since it cannot compensate for the lack of information due to under-sampling. Rather, it is aimed at improving the detection and/or the localization of the scattering targets by reducing ambiguities due to aliasing replicas.

In order to detail the multi-frequency fusion procedure, it is convenient to start from the one-dimensional version of Equation (4) and omit the unessential factors outside the integral,

$$E(u) = \int_{-\chi_{in}}^{\chi_{in}} J_{eq}(\chi) \exp{(j2\pi u\chi)}d\chi \qquad (10)$$

where $u = \sin\theta$, $\chi_{in} = x_{in}/\lambda$, and $\lambda = c/f$, $f$ being the operating frequency and $c$ the speed of light. According to the Nyquist criterion, to reconstruct a current supported over $\chi \in [-\chi_{in}, \chi_{in}]$ the field must be sampled with the step $\Delta u = 1/(2\chi_{in})$. Assume that $J_{eq}(\chi) = \delta(\chi) \exp{j\phi(\chi)}$ and

that the field is known within the interval $u \in [-u_{max}, u_{max}]$; this basically entails studying the point-spread-function (PSF) of the reconstruction procedure. In addition, assume that $N$ samples are available at the step $\Delta u = 2u_{max}/N$. Accordingly, in the Fourier domain, the model data can be written as

$$E(u) = \text{rect}[u/(2u_{max}))] \exp j\phi(0) \sum_i \delta(u - i\Delta u) \tag{11}$$

and the corresponding image writes as

$$|J_{eq}(\chi)| = |2u_{max} \sum_i \text{sinc}[2\pi u_{max}(\chi - i/\Delta u)]| \simeq$$

$$\simeq 2u_{max} \sum_i |\text{sinc}[2\pi u_{max}(\chi - i/\Delta u)]| \tag{12}$$

where $\text{sinc}(x) = \sin(x)/x$ and in the last term we exploited the fact that the main beams of the different replicas do not overlap. On making explicit the frequency dependence, one obtains

$$|J_{eq}(x, f)| \simeq 2u_{max} \sum_i |\text{sinc}[\pi(2u_{max}xf/c - iN)]| \tag{13}$$

As can be seen, the reconstruction consists of a train of sinc pulses centered at $\Delta x = \lambda N/(2u_{max})$ with the main lobe $2l = \lambda/u_{max}$ large (see Figure 5).

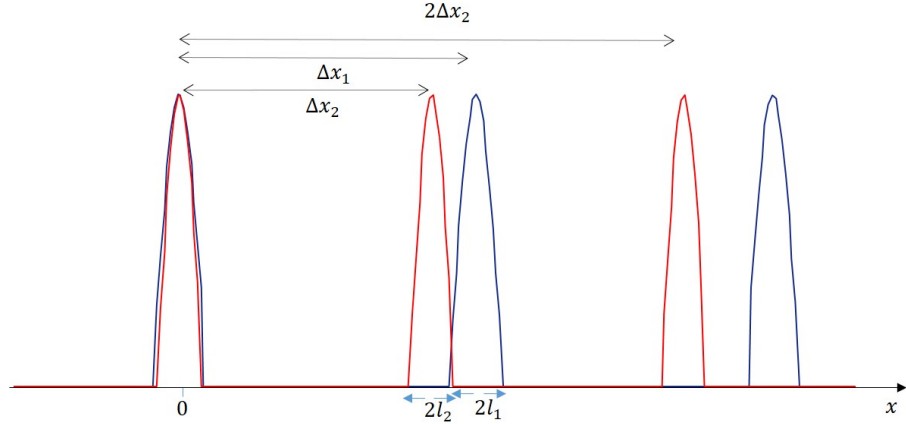

**Figure 5.** Sketch of main lobe, first and second replicas at different frequencies $f_1$ (blue) and $f_2 > f_1$ (red).

Consider now the reconstructions obtained at two different frequencies, say $f_1$ and $f_2 > f_1$. The reconstructions now consist of sinc pulses of different width, say $2l_i$ with $i \in (1, 2)$, both have a peak at the source position $x = 0$ but the replicas appear at different rates. This simple framework suggests the adoption of a multiplicative fusion scheme to mitigate artifacts. Indeed, it is apparent that multiplying $|J_{eq}(x, f_1)|$ and $|J_{eq}(x, f_2)|$ still retains the peak at $x = 0$ (i.e., the actual source position) where the two contributions overlap, while other peaks may be reduced with a proper choice of the frequency step. In particular, in order to completely avoid the first replicas of $|J_{eq}(x, f_1)|$ and $|J_{eq}(x, f_2)|$ overlap, it must hold that

$$\Delta x_1 - \Delta x_2 \geq l_1 + l_2 \tag{14}$$

i.e.,

$$\lambda_1 \frac{N}{2u_{max}} - \lambda_2 \frac{N}{2u_{max}} \geq \frac{\lambda_1}{2u_{max}} + \frac{\lambda_2}{2u_{max}} \tag{15}$$

which leads to the condition

$$f_2 \geq f_1 \frac{N+1}{N-1} \tag{16}$$

which in turn dictates how to choose the subsequent higher frequency. In the same way, if we want that the second replica at $f_2$ does not overlap with the first one at $f_1$, the condition is

$$2\Delta x_2 - \Delta x_1 \geq l_1 + l_2 \tag{17}$$

that leads to the condition

$$f_2 \leq f_1 \frac{2N - 1}{N + 1} \tag{18}$$

which can be satisfied when $N > 5$.

How must conditions in Equations (16) and (18) be interpreted and what do they guarantee about artifacts in the reconstruction?

Equation (16) basically represents the frequency step, so that the frequencies can be chosen as

$$f_{n+1} \geq f_n \frac{N + 1}{N - 1} \tag{19}$$

whereas Equation (18) sets somehow the frequency band, because once $f_1$ has been fixed,

$$f_n \leq f_1 \frac{2N - 1}{N + 1} \tag{20}$$

If Equations (19) and (20) are both fulfilled, then an artifact-free region of about $[-\Delta x_1, \Delta x_1]$ is guaranteed around the target. What is more, previous criteria depend on the number of data $N$ and, hence, points out how spatial under-sampling can be somehow and partially counteracted by collecting data at different frequencies. In particular, a lower $N$ requires choosing subsequent frequencies at higher multiplicative steps, which roughly can be thought of as a trade-off between data reduction and frequency bandwidth requirements.

In order to more easily get the required bandwidth, condition (Equation (14)) can be relaxed by accepting a partial superposition of the aliasing replicas. More in detail, by setting $\Delta x_1 - \Delta x_2 \geq \alpha(l_1 + l_2)$ with $\alpha < 1$, the subsequent frequencies can be chosen according to the criterion

$$f_n \geq f_{n-1}(N + \alpha)/(N - \alpha) \tag{21}$$

Note that if a larger free of artifacts region should be required, the same reasoning as above can be followed to derive different, and generally more stringent, conditions. Finally, we stress that similar arguments can be applied for the other spatial dimension. Eventually, one adopts the more severe requirements, among the two, so that for both dimensions, the non-overlapping criteria are satisfied.

According to previous results, the detection and localization of the buried target can be achieved by the following fusion strategy

$$I(x, y) = \prod_n \|J_{eq}(x, y, f_n)\| \tag{22}$$

where $J_{eq}(x, y, f_n)$ is obtained by a standard linear inversion scheme based on the truncated singular values decomposition procedure and $\|J_{eq}(x, y, f_n)\| = \sqrt{|J_{xeq}(x, y, f_n)|^2 + |J_{yeq}(x, y, f_n)|^2}$ accounts for both the reconstructed surface current components.

It is worth noting that the choice of a multiplication based fusion strategy, of course is not the only way to merge the different reconstructions. Indeed, the optimization of this point is beyond the scope of the presented paper as the focus is on the possibility of exploiting non-coherent reconstructions to mitigate spatial under-sampling. Moreover, it must be pointed out that even though the proposed multiplicative procedure can impact positively on the achievable resolution and the side-lobe level, the result in Equation (22) is, in general, no longer clearly linked to the targets' shapes. In this regard, Equation (22) must be considered more as an indicator function which highlights the presence of targets. This is, however, sufficient for detection, with the latter being the main aim of this contribution.

## 5. Numerical Examples

In order to check the proposed approach, in this section, we report some numerical examples.

We start by considering first the targets embedded in free-space. This, of course is the simplest scenario against which to check the detection and the localization ability of the method and represents a benchmark for the more interesting half-space background medium.

The incident field is provided by a $y$ polarized plane-wave impinging along $\theta = \pi$ direction, the equivalent (reconstruction) plane is considered at $z_p \leq 0$ that can be varied in order to explore the depth as well. The investigation domain is a square of side 4 m (that is $x_{in} = y_{in} = 2$ m ) whereas rectangular metallic plates are considered as targets. Finally, the scattered field is observed over the observation domain $\theta \leq \theta_{max} = \pi/4$. For this simple case, the scattered field is computed by adopting a standard method of moment based on the Physical Optics approximation. However, all the data are corrupted by additive white noise with $10dB$ of signal-to-noise ratio (SNR).

In order to perform the single-frequency reconstruction, first, the linear integral operator Equation (1) is discretized by a standard method of moment [26]. To this end, here, Fourier harmonics are used to expand the unknown source function and a point matching procedure is set in the data space. Thus, the reconstruction is obtained by a truncated singular values decomposition procedure. Note that, even though both $J_{xeq}(x, y, f)$ and $J_{yeq}(x, y, f)$ are reconstructed (to get the reconstruction $\|J_{eq}(x, y, f)\|$), according to the presented model, the singular value decomposition needs to be computed only once. Moreover, this holds true even when reconstructions at different depths are desired. Indeed, in this case it is sufficient to change the $z_p$ parameter in Equation (4), correspondingly normalize the scattered field, and rerun the reconstruction procedure.

We start by considering two metallic plates of $0.30 \times 0.15$ m$^2$. The first one is centered at $(-0.5, -0.1, -0.2)$ m (center point) whereas the second one at $(0.6, 0.1, -0.2)$ m. Hence, the two targets are at the same depth. For the configuration described above, the results in Section 3 say that, at $f = 1GHz$, more than 200 observation points would be needed. In order to have a benchmark, Figure 6 shows the normalized single frequency reconstruction, $\|J_{eq}(x, y, f)\|$ (with $f = 1$ GHz), of the two plates obtained by using full data, i.e., by sampling the $u - v$ domain as dictated by Section 3. In particular, we fixed $N = 15$, so that the overall observations are $N^2$ and set $z_p = 0$. As can be seen, both the targets are clearly detected and localized.

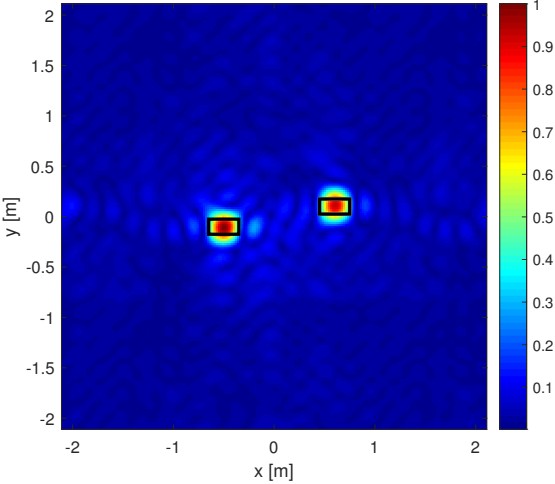

**Figure 6.** Normalized linear scale) single-frequency reconstruction of two metallic rectangular plates both located at the same depth of $-0.2$ m. Full data (i.e., $15 \times 15$ observations evenly taken in the $u - v$ plane) at $f = 1$ GHz have been used and $z_p = 0$. Black solid line denotes actual plates' shape and position.

Now, we switch to considering under-sampled data. To this end, the observations are reduced to $4 \times 5 = 20$ (still evenly taken in the $u - v$ plane), that is, less than $1/10$ of the required ones. The corresponding reconstruction is shown in Figure 7. As expected, the reconstruction is corrupted by many aliasing artifacts that make it impossible to discern where the targets actually are. To overcome this drawback, we use the multi-frequency approach. In particular, here we choose $\alpha = 0.4$ in Equation (21) so that the corresponding frequencies turn out to be $(0.6694, 0.8182, 1.0000, 1.2222, 1.4938)$ GHz. Note that we kept $f = 1$ GHz as the central frequency. The corresponding multi-frequency reconstruction $I(x, y)$ is shown in Figure 8. Now, the image is much less cluttered with artifacts than Figure 7 and the two targets are clearly detected and accurately located. One may also note that $I(x, y)$ fails to reproduce the size (even only roughly) of the targets. This could be expected by its very definition. Indeed, $I(x, y)$ is an indicator function which should reveal whether there is a target in the scene.

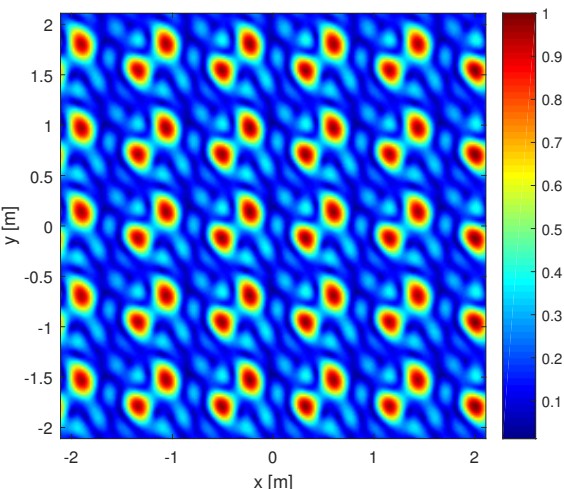

**Figure 7.** Normalized linear scale) single-frequency reconstruction for the same case as Figure 6 but with under-sampled data taken at only $4 \times 5$ observation points at $f = 1$ GHz.

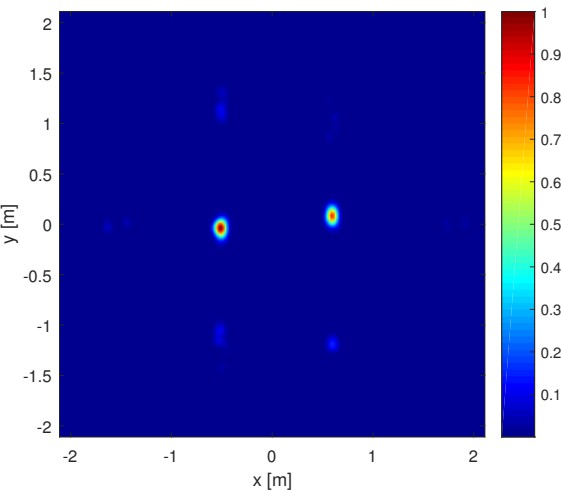

**Figure 8.** Normalized (linear scale) multi-frequency reconstruction for the same case as Figure 7 using the same under-sampled observations but five frequencies $(0.6694, 0.8182, 1.0000, 1.2222, 1.4938)$ GHz.

We now take a little step ahead and consider a more involved scattering scenario. To this end, we consider the two metallic plates being of different sizes and orientations and located at different

depths. The first target is a rectangular plate of $0.30 \times 0.15$ m$^2$ located at $(0.6, 0.1, -0.1)$ m. The second one is located at $(-0.5, -0.1, -0.3)$ m, $0.45 \times 0.30$ m$^2$ in size and rotated of $-\pi/4$. Figures 9–12 show the corresponding multi-frequency reconstructions for the same measurement configuration as Figure 8, except for $z_p$ that is varied from 0 m to 0.3 m. All the figures appear rather weakly corrupted by aliasing artifacts. However, the two targets are not always detected. For example, by looking at Figure 9 (i.e., the one obtained by setting $z_p = 0$ m) only the target located at $z = -0.1$ m is detected and correctly located. This is because this target is closer to the reference plane located at $z_p$. However, by increasing the depth of the reference plane the second target starts to be detect (see Figure 10 where $z_p = -0.1$ cm) and becomes clearly detected and well localized when $z_p$ reaches $-0.3$ m. By contrast, when $z_p$ increases, the first target becomes undetected. This example is quite interesting since it proves that targets located at different depth can be actually detected by changing the reference depth $z_p$. What is more, as described in the previous section, this can be achieved with a minor processing cost.

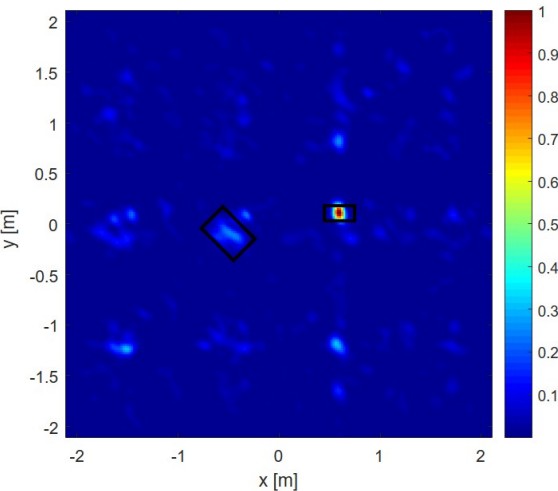

**Figure 9.** Normalized (linear scale) multi-frequency reconstruction of two metallic plates located at different depths and having different sizes and orientations. The parameter of configuration are the same as Figure 8. Black solid line denotes actual plates' shape and position.

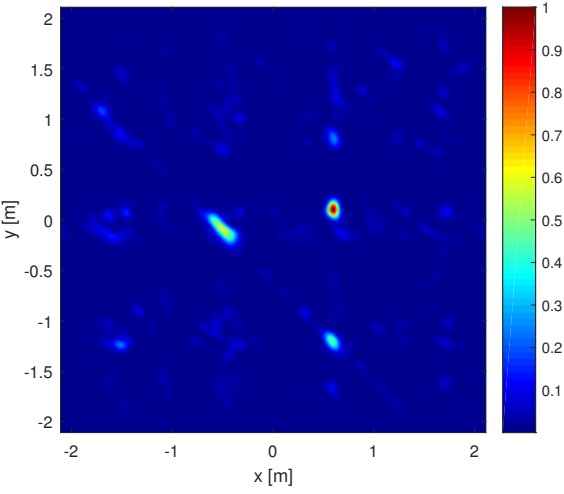

**Figure 10.** The same as Figure 9 but with $z_p = -0.1$ m.

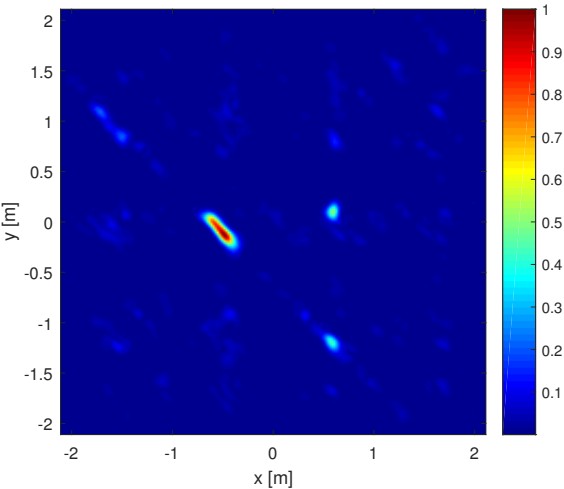

**Figure 11.** The same as Figure 9 but with $z_p = -0.2$ m.

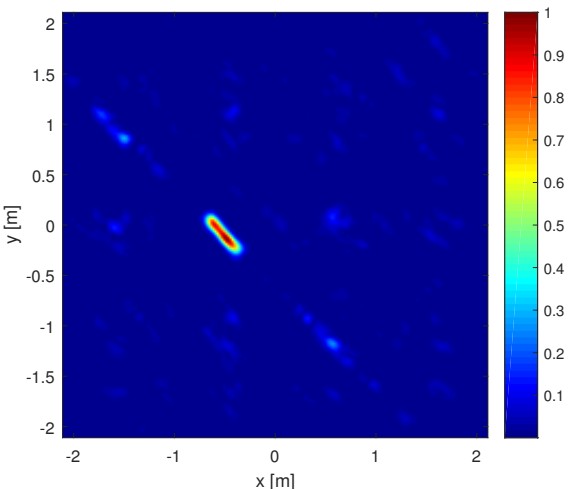

**Figure 12.** The same as Figure 9 but with $z_p = -0.3$ m.

As a further example, we reconsider the two targets of the latter case embedded within a half-space having relative dielectric permittivity 4 and conductivity 0.02 S/m. In this case, the scattered field data have been generated by a commercial forward solver whereas the parameters of the configuration are maintained the same as in the previous examples. Some reconstructions corresponding to this case are reported in Figures 13 and 14. As can be seen, the two targets are detected and well localized transversely. As to the depth, the same behaviour as above is observed, that is, the reconstruction is stronger for that target which is *closer* to the reference plane. Of course, since the half-space medium features are generally unknown, the estimated depths will not coincide with the actual ones.

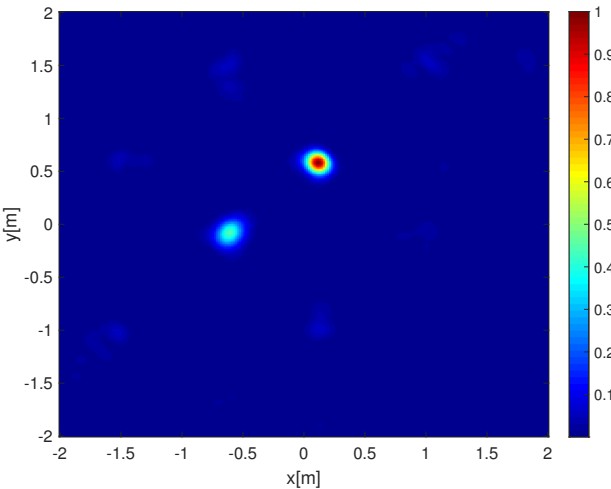

**Figure 13.** Normalized (linear scale) multi-frequency reconstruction of two metallic plates considered in Figure 9 but now buried within a half-space. The parameter of configuration are the same as Figure 8, the considered depth is $z_p = -0.1$ m.

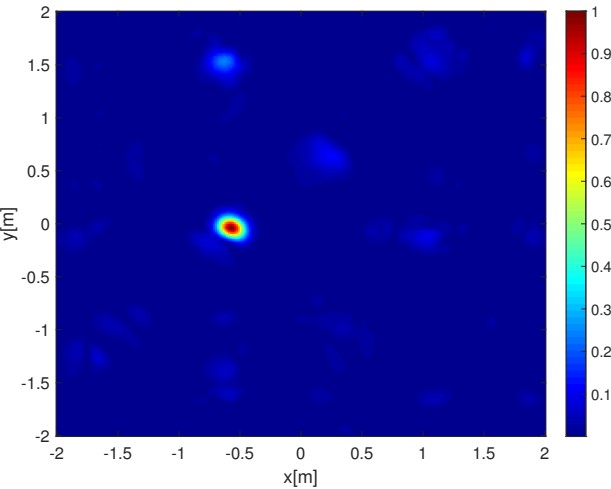

**Figure 14.** The same as Figure 13 but with $z_p = -0.5$ m.

The concluding example is shown in Figures 15 and 16. In particular, in order to check the method against targets which are not plates, in those figures cylindrical targets have been considered. The first target is a cylinder having radius of 0.085 m, length of 0.40 m and is centered at $(-0.35, -0.35, -0.2)$ m; the second cylinder has radius of 0.075 m, length of 0.30 m and is centered at $(0.3, 0.3, -0.1)$ m. The location and orientation of the target is reported in Figure 15. As can be seen, despite the very few employed measurements, in this case, the method clearly detects the two targets. As expected, the target which is closer to the interface *appears first*, whereas the other one becomes more evident when the depth is decreased: once again exploring depth revealed a crucial and distinctive feature of the method. Finally, it is observed that one of the target reconstruction exhibits two maxima. This may be due to multiple reflections between the air/soil interface and the target. In the time domain, these amultiple bounces manifest as more deeply located "spurious" targets. At single-frequency, instead, multiple bounces impact on the (complex) amplitude of the current induced on the target and, hence, of the equivalent current. Therefore, depending on the configuration parameters, and in

particular on the distance between the target and the interface, multiple reflections can shape the current amplitude.

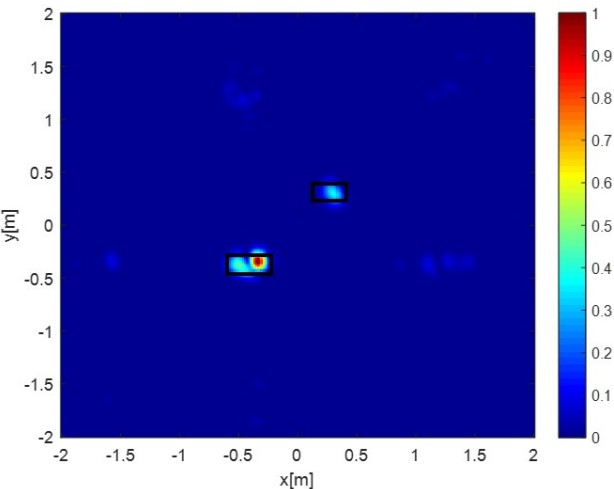

**Figure 15.** Normalized (linear scale) multi-frequency reconstruction of two metallic cylinders with axis parallel to the *x*-axis, buried within a half-space. Black solid line denotes actual cylinders' position. The parameter of configuration are the same as Figure 8, the considered depth is $z_p = -0.1$ m.

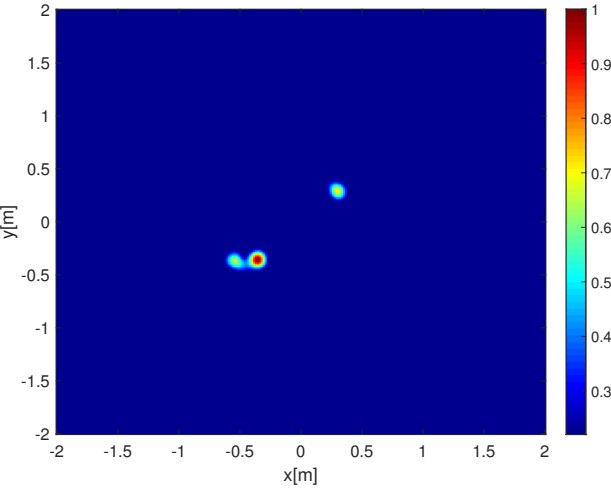

**Figure 16.** The same as Figure 15 but with $z_p = -0.4$ m.

## 6. Conclusions

A method for detecting and localizing shallowly buried targets from under-sampled multi-frequency far-zone scattered field data has been presented. Basically, the method relies on two main ingredients: a suitable scattering model and a simple procedure to process multi-frequency under-sampled data.

The scattering model is derived according to the equivalence theorem which allows to cast the detection as the reconstruction of equivalent sources supported over a reference plane (above the targets). Inverting this model offers a number of advantages. First, the detection becomes a $2D$ problem with obvious reduction of the computational burden. Also, since the incident field is embodied into the unknown equivalent sources, coherence (i.e., synchronization) between the TXs and Rxs is not necessary when single-frequency data are employed. This in principle makes the overall system more cost effective.

Even under this model and single-frequency data the study of the NDF of the scattered field has revealed that the number of far-zone spatial measurements required to correctly image the equivalent currents may become impractically large, especially for electrically large investigation domains. In order to reduce the need for such a high number of spatial measurements (and, hence, the system cost) we have proposed to use multi-frequency data. In particular, here frequencies are used to mitigate artifacts that arise when data are reduced with respect to the NDF. Since artifacts are frequency dependent, images obtained at different frequencies are fused via a simple multiplicative strategy. A criterion to choose the frequencies in order to have a spatial region free from artifacts has been provided as well.

A drawback of the proposed method is that the target's depth is not *directly* retrieved. However, this appears a minor drawback when the targets are shallowly buried beneath the air/soil interface (this is just the case we focused on herein). Moreover, by changing the reference depth $z_p$ the proposed method exhibits a certain ability to retrieve the targets' depths with a minor impact on the computational cost.

Numerical results for scattering objects embedded in free-space and within a half-space medium confirmed the feasibility of the approach. The presented results appear promising but represent only a preliminary step towards the experimental validation of the method, since on-field conditions will present harder challenges to be faced. However, the proposed strategy is linear and intrinsically very robust and stable against uncertainties on data; hence, we are confident that our method will work also for realistic scenarios. Also, we have to mention that we already made some experiments for the case of a homogeneous medium (targets were not buried) as reported in [26]. Actually, to test the proposed strategy against experimental data for a subsurface scattering is an our commitment that will be addressed in future developments.

Finally, we observe that it would be interesting to compare the proposed method with others present in the literature. Most of those methods employ multi-frequency coherent processing, hence the comparison with them would not be fair. On the other hand, comparison with a single-frequency (full-data) method is already contained in this paper. To our knowledge, a fair comparison could be achieved only against compressed sensing methods [37], which can work with a reduced number of measurements and by considering one frequency at time.

**Author Contributions:** Each author contributed extensively to the preparation of this manuscript. In particular, conceptualization, A.B., G.L. and R.S.; methodology, A.B., G.L. and R.S.; software, A.B., A.D. and R.S.; validation, A.B., A.D. and R.S.; writing—original draft preparation, A.B. and R.S.; writing—review and editing, A.B., A.D., G.L. and R.S.

**Funding:** This research was funded by Ministero dello Sviluppo Economico, within the program UE-FESR, PON "Imprese e competitivitá" 2014–2020, under the project "IDROS" (Impiego di Droni per la Ricerca nel sottOSuolo) grant number F/050187/02/X32.

**Conflicts of Interest:** The authors declare no conflict of interest.

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
