# Peer review of "Subsurface Detection of Shallow Targets by Undersampled Multifrequency Data and a Non-Cooperative Source"

_applsci, doi:10.3390/app9245383_

Round 1

Reviewer 1 Report

Some suggestions:

Line 7) In the abstract the authors could to specify a little bit wat is the equivalence theorem; for example: “the equivalence theorem for electromagnetic radiation”… they are others equivalence theorems

Line 8) The same brief description of a non-cooperative source

Line 17) Best: “Shallow subsurface prospecting.   In Keywods

Lines19-23) The yellow market sentence is risky because the authors are confusing the phenomenon of the reflection of electromagnetic waves with the phenomenon of scattering. They have put the bibliographic citations mixing these two concepts.... they should review this

Lines 33-34) “Large measurement apertures” remain a little ambiguous, because "aperture" is not used when describing the geometry of the data acquisition and does not agree with the “high transverse resolution”

Lines 47-48) “Blind” is not clear. It means signal saturated or lower signal?... Clarify this point because it is used later.

Line 49) The [26] reference need a previous explanation

Line 105) The reference [30] is missing information

Section 4) In this section, in lines 240-261 (approx.), the authors the authors could introduce the concept that the scatering radiation frequency is usually less than the illumination frequency of the source, so the equations presented must meet this condition.

Figures 6 to16 are displaced respect to the text.

Reviewer 2 Report

The paper is well-written with the results well-presented.

Only one minor comment. In the paper the authors only tested their methodology using numerical examples. It will be interesting to test the proposed multi-frequency technique using experimental data as other challenges might arise.
